# Soft Tissue Grafting Procedures before Restorations in the Esthetic Zone: A Minimally Invasive Interdisciplinary Case Report

**DOI:** 10.3390/medicina59050822

**Published:** 2023-04-23

**Authors:** Gerardo Guzman-Perez, Carlos Alberto Jurado, Francisco X. Azpiazu-Flores, Humberto Munoz-Luna, Kelvin I. Afrashtehfar, Hamid Nurrohman

**Affiliations:** 1Department of Graduate Periodontics, Multidisciplinary Educational Center in Oral Rehabilitation (CEMRO), Tarímbaro 58893, Mexico; 2Department of Prosthodontics, The University of Iowa College of Dentistry and Dental Clinics, Iowa City, IA 52242, USA; 3Department of Restorative Dentistry, Gerald Niznick College of Dentistry, University of Manitoba, Winnipeg, MB R3E 3N4, Canada; 4Private Dental Technician, Irapuato 36643, Mexico; 5Division of Restorative Dental Sciences, Clinical Sciences Department, Ajman College of Dentistry, Ajman City P.O. Box 346, United Arab Emirates; 6Department of Reconstructive Dentistry and Gerodontology, School of Dental Medicine, University of Bern, 3010 Bern, Switzerland; 7Missouri School of Dentistry & Oral Health, A.T. Still University, Kirksville, MO 63501, USA

**Keywords:** dental esthetics, restorative dentistry, permanent dental restoration, dental ceramics, treatment outcome, patient care team, soft tissue surgical procedures, periodontal plastic surgery

## Abstract

An esthetically pleasing smile is a valuable aspect of physical appearance and plays a significant role in social interaction. Achieving the perfect balance between extraoral and intraoral tissues is essential for a harmonious and attractive smile. However, certain intraoral deficiencies, such as non-carious cervical lesions and gingival recession, can severely compromise the overall aesthetics, particularly in the anterior zone. Addressing such conditions requires careful planning and meticulous execution of both surgical and restorative procedures. This interdisciplinary clinical report presents a complex case of a patient with esthetic complaints related to asymmetric anterior gingival architecture and severely discolored and eroded maxillary anterior teeth. The patient was treated using a combination of minimally invasive ceramic veneers and plastic mucogingival surgery, resulting in a successful outcome. The report emphasizes the potential of this approach in achieving optimal esthetic results in challenging cases, highlighting the importance of an interdisciplinary team approach in achieving a harmonious balance between dental and soft tissue aesthetics.

## 1. Introduction

Dental esthetics play a crucial role in modern society, as an esthetic smile is associated with kindness, popularity, intelligence, and high social status [1]. Achieving an aesthetically pleasing smile requires a harmonious balance between various intraoral and extraoral elements, including the lips, teeth, and gingival tissues [2,3,4,5,6]. Gingival esthetics, also known as “pink esthetics”, is a critical determinant of overall dental esthetics, as they frame any esthetic restorative work performed on the teeth [1]. Therefore, conditions such as gingival recession can significantly compromise the esthetic outcome, especially when multiple anterior maxillary teeth are involved [7,8]. Over the years, several surgical procedures have been proposed to manage these complex cases, including coronally repositioned flaps, lateral sliding flaps, free gingival grafts, and subepithelial connective tissue grafts using tunneling and envelope flap techniques [7,8,9,10,11,12]. These procedures aim to establish a harmonious gingival architecture, increase the amount of keratinized gingiva, and reduce hypersensitivity related to denuded root surfaces [13]. However, to achieve satisfactory results, precise surgical techniques must be employed and attention to detail is necessary to prevent the formation of post-surgical defects, such as scarring, keloid-like defects, and loss of the interdental papilla [7,9]. Data have shown that the outcomes of the surgical treatment of gingival recession are positive. A recent Cochrane systematic review evaluating root coverage procedures for single and multiple recession has reviewed 48 randomized controlled trials in which gingival recessions were addressed with subepithelial connective tissue grafts plus coronally advanced flap and tissue guided regeneration with resorbable membranes plus connective tissue grafts and it concluded that all procedures can be successfully provided for treating single or multiple gingival recessions [14].

Ceramic dental laminate veneers are a well-documented and predictable treatment for modifying the shade of whitening-resistant teeth, improving the shape of teeth with acquired malformations and loss of facial enamel, correcting minor rotations, and closing moderate diastemas [15,16,17,18]. Contemporary dental ceramic systems have improved mechanical and optical properties [19,20,21]. When ceramic restorations are conditioned and combined with resin-based cement, a strong and long-lasting chemical, the micromechanical bond can be created between the ceramic and the underlying tooth substrate [22,23]. Despite significant advances in adhesive dentistry, the success of treatment with dental veneers depends on several factors, including thorough planning, adequate preparation and conditioning of the tooth structure, satisfactory design and conditioning of the restorations, and preparation of the receptor dental and gingival tissues to ensure the harmonious integration of the restorations with the rest of the mouth [24,25]. This clinical report presents the interdisciplinary management of a patient with esthetic complaints related to asymmetric anterior gingival architecture and severely discolored and eroded maxillary anterior teeth. The patient was treated with a combination of minimally invasive ceramic laminate veneers and plastic mucogingival surgery.

## 2. Materials and Methods

A 32-year-old male patient presented to the clinic indicating that he did not like the esthetics of his anterior teeth (Figure 1A,B).

After a medical questionary, the patient was classified as an ASA type 1 patient systematically healthy with no medical conditions to be concerned about prior to dental care. Additionally, he expressed suffering from severe root sensitivity when drinking or eating. At the initial consultation, the patient expressed that he used to chew on citrus fruits with his anterior teeth and that lemon juice was a big part of his diet until he noticed defects in his anterior teeth. Intraorally, generalized loss of clinical attachment and Miller type 1 gingival recession was observed in the maxillary right first premolar, right canine, and both maxillary central teeth. Additionally, cervical non-carious lesions were noticed on teeth maxillary right first premolar, right canine, right lateral, right central, and maxillary left central, and significant loss of facial enamel was noticed on both maxillary central teeth, giving the teeth a lower color value than the adjacent anterior teeth (Figure 2A,B).

When the endodontic condition of the anterior teeth was assessed, all the teeth responded positively to electrical and thermal tests. Additionally, no signs of parafunction or occlusal interferences were noticed during the examination. Subsequently, the clinical findings were presented to the patient and three treatment options were offered: (1) no treatment provided with only monitoring of the gingival conditions and address whenever patient decides; (2) only restorative prosthesis such as veneers from canine to canine to improve the shade and shape of the anterior dentition; and (3) soft tissue grafting procedure to treat the gingival recession following with veneers from canine to canine.

The patient was more interested in the third option and it was further explained that it would consist of plastic mucogingival surgery involving tunneling connective tissue grafts, and anterior ceramic veneers were presented. The treatment’s limitations, possible complications, length, and financial aspects were discussed; after considering all these factors, the patient decided to proceed with the proposed treatment.

On the day of the surgery, bilateral anterior superior alveolar, nasopalatine, and left greater palatine blocks were performed with Lidocaine 2% with 100,000 units of epinephrine (Lignospan Standard; Septodont, Saint-Maurdes-Fosses, France). Subsequently, the tunneling procedure was performed from maxillary right first premolar to maxillary left lateral tooth; a partial thickness dissection was performed using periodontal microsurgical instrumentation. Additionally, the root surfaces were mechanically debrided and the smear layer was chemically removed using 24% ethylenediaminetetraacetic acid (EDTA) gel (Straumann PrefGel; Straumann, Basel, Switzerland) for 2 min, followed by abundant irrigation with saline solution (Figure 3A,B).

Subsequently, a partial thickness flap was reflected approximately 4–5 mm from the lingual marginal gingiva of the maxillary right second molar, extending to the lateral tooth in the same quadrant. A 55 × 5 mm band of connective tissue was obtained from the glandular and adipose regions of the palate. During the harvesting procedure, absolute care was taken to preserve the periosteum of the donor site. Subsequently, the connective tissue graft was trimmed to fit the receptor sites (Figure 4A,B) and was inserted and secured below the partial thickness anterior flaps apical to the cementoenamel junction of the anterior teeth with 5-0 monofilament nylon sutures (5-0 Ethilon; Ethicon, Raritan, New Jersey, NJ, USA) (Figure 4C–E).

Additional vertical suspensory extending from the base of each interdental papilla to coronal anchorage points were created using flowable composite (3M Filtek Supreme Flowable Composite; 3M ESPE, St Paul, MN, USA). The suspensory sutures were carefully placed to secure the gingival graft, maximize graft coverage, and maintain the height of the interdental papillae (Figure 5A,B).

A home maintenance regime consisting of a soft diet, careful cleaning of the sites with a soft bristle brush (GUM Post-Surgical Toothbrush; Sunstar Americas, Schaumburg, IL, USA), and oral rinses with 0.12% Chlorhexidine Gluconate (Peridex Oral Rinse; 3M ESPE, St Paul, MN, USA) twice a day for 7 days was established after surgery.

The surgical sites were reevaluated 48 h, 2 weeks, and 1 month after surgery. At this stage, satisfactory root coverage and adequate healing with complete reepithelization of the donor site were observed clinically (Figure 6A–C).

Additionally, the patient denied any discomfort or complication during healing related to the surgical procedures performed. After 6 months of healing, the tissues were reassessed and deemed adequate to restore the anterior teeth with ceramic veneers (Figure 7A,B).

An additive diagnostic wax-up of the anterior teeth was performed and used as a reference for fabricating a reduction guide using heavy-bodied condensation silicone (Zetaplus, Putty Zhermack; Rome, Italy). Subsequently, the labial erosive lesion on the maxillary left lateral tooth was blocked out using resin-modified glass ionomer cement (GC Fuji 2: GC America, Alsip, IL, USA), and conservative laminate veneer preparations extending 0.2 below the free gingival margin were performed. Using the reduction guide as a reference, careful attention was paid to keep the preparation on enamel, ensuring a reduction of approximately 0.5 mm in the labial surfaces, anterior line angles, and incisal edges (Figure 8A–C). The patient was offered either CAD/CAM or traditional hand-made intraorally made provisionals, and due to costs the patient selected traditional provisional restorations. Temporary restorations were fabricated with bisacryl-based composite resin (ProTemp Plus; 3M ESPE, St Paul, MN, USA) and retained using the spot-etch technique [17] (Figure 8D).

Subsequently, impressions were made with irreversible hydrocolloid (Geltrate, Dentsply Sirona, Fair Lawn, NJ, USA) and diagnostic casts were fabricated with type III dental stone (Buff Stone; Whip Mix Corp, Louisville, KY, USA). Using the contours of the diagnostic casts as reference, 6 lithium disilicate (E.max CAD; Ivoclar Vivadent, Schaan, Liechtenstein, Switzerland) ceramic veneers were fabricated using a professional computer-aided design and computer-aided manufacturing (CAD-CAM) dental software (Exocad, Exocad GmbH; Darmstadt, Germany) (Figure 8A,B). Subsequently, the restorations were tried intraorally and the interproximal contact and margins were assessed and carefully adjusted. After complete seating was achieved, the restorations were etched for 20 s with hydrofluoric acid (IPS Ceramic Etching Gel; Ivoclar Vivadent, Schaan, Liechtenstein, Switzerland), primed (MonoBond Plus; Ivoclar Vivadent, Schaan, Liechtenstein, Switzerland), and cemented with resin-based cement (MultiLink Automix; Ivoclar Vivadent, Schaan, Liechtenstein, Switzerland) (Figure 9A,B).

The patient was provided with instructions for proper maintenance including brushing teeth with a soft-bristled toothbrush thoroughly twice a day, flossing daily between teeth to remove food and plaque, and dental cleanings twice a year. The patient was reassessed 1 week, 1 month, and 6 months after delivery. During the 1-year follow-up appointments, the patient denied any discomfort or issues related to the restorations provided (Figure 10A,B).

## 3. Results

The case study describes a 32-year-old male patient who presented to the clinic with esthetic concerns regarding his anterior teeth and severe root sensitivity. Generalized clinical attachment loss and gingival recessions were observed, along with cervical non-carious lesions and significant loss of facial enamel. The endodontic condition of the teeth was assessed and found to be positive. The patient was presented with a treatment plan consisting of plastic mucogingival surgery involving tunneling connective tissue grafts and anterior ceramic laminate veneers. The patient agreed to proceed with the proposed treatment. A unilateral (left) side was used for the donor site and the amount was able to cover the receptor sites. Bilateral anterior superior alveolar, nasopalatine, and left greater palatine blocks were performed, followed by tunneling and connective tissue grafting. The surgical sites were reevaluated at different intervals and adequate root coverage and healing were observed clinically. Conservative veneer preparations were performed using a reduction guide as a reference and the patient was re-assessed after delivery and during a 1-year follow-up period with no reported issues.

## 4. Discussion

An attractive smile plays a significant role in contemporary society [1], and with the advances in dental material science and CAD-CAM technologies, highly esthetic ceramic restorations [26,27,28] can now be designed and manufactured consistently and predictably [18]. To achieve satisfactory results, gingival esthetics must be established to provide a pleasing “frame” for dental restorations [1]. Successful plastic surgery periodontal procedures have been used to correct deficiencies related to gingival recession [11,13]. Connective tissue grafts have been reported to provide satisfactory root surface coverage, ranging from 76.6% to 100%, depending on the size and configuration of the recession, the location, and the number of teeth treated [7,9]. Research suggests that connective tissue grafts and tunneling techniques offer advantages over other periodontal procedures, such as coronally and horizontally repositioned flaps, as the forces exerted by intraoral muscle insertions can be circumvented [7] and the integrity of fragile interproximal tissues can be preserved [9]. Moreover, this treatment modality permits predictably enhancing the amount of keratinized tissue available, thus creating a more maintainable periodontal environment [8]. Both elements are crucial to ensure the satisfactory esthetic integration of restorations with equigingival or subgingival finish lines [14]. Previous case reports treating multiple recessions in the esthetic zone have shown positive outcomes. Case series treating 22 patients with a total of 73 recessions with a mean depth of 2.8 mm were treated with the coronally advanced flap technique and reevaluated at 1 year, resulting in an average of 97% root surface coverage, and a complete root coverage was achieved in 16 out of the 22 patients [29]. Patients were treated with a similar approach in another case series with a longer follow-up treating 22 patients with 73 gingival recessions in the esthetic zone Miller Class I and II [30]. The results displayed 94% of root coverage at 5 years of examination and complete coverage was obtained in 15 out of the 22 patients [30]. Moreover, a systematic review from the American Academy of Periodontology (AAP) evaluated the success of soft tissue root coverage procedures. This review evaluated 234 clinical trials on class I, II III, and IV gingival recessions, and it demonstrated that all clinical procedures provided a significant reduction in recession depth, concluding that connective tissue grafts provide the best outcome for clinical practice [31]. Due to the positive clinical reports, soft tissue grafting procedures can be considered in the anterior region to fulfill the patient’s esthetic demands.

Traditionally, root surfaces are cleaned with compounds such as citric acid or tetracycline hydrochloride preceding root coverage procedures to remove the smear layer, open the dentinal tubules, and decrease the microbial load and bacterial cytotoxic byproducts [7,8,9]. In the present clinical report, 24% EDTA gel (Straumann PrefGel; Straumann, Basel, Switzerland) was applied for 2 min before the connective tissue graft application. Ethylenediaminetetraacetic acid is an organic compound widely used in endodontics as an adjuvant irrigation agent due to its effectiveness in removing calcium by means of chelation [18]. Highly concentrated gel preparations of this compound are advantageous since their viscosity permits the selective treatment of the intended surfaces only, thus avoiding the highly concentrated formulation’s unnecessary demineralization of other dental tissues and substances. Some studies have recommended the results obtained with EDTA gel. A clinical trial evaluating EDTA gel conditioning during periodontitis assessed the surface topography and periodontal ligament cell adhesion with and without the application of EDTA gel, and the results concluded that it provides the most desirable root surface to which maximum periodontal ligament cells can adhere and on which they can grow [32].

Ceramic restorations are a well-supported treatment that can improve the patient’s self-esteem and social interactions [1,14]. Modern dental ceramics possess mechanical and chemical properties, making them excellent restorative materials. Feldspathic and glass-infiltrated ceramics can be bonded to the enamel using an adhesive hybrid layer [18] and have a modulus of elasticity similar to the enamel, making them a suitable “biomimetic” alternative to replace lost enamel [14]. In the present clinical report, the teeth were minimally prepared using the desired restorative contours established with a diagnostic wax-up as a reference. This step was critical to ensure that the tooth preparations preserved as much enamel as possible to guarantee adequate union between the restoration and the substrate, and to provide enough space to fabricate esthetic and structurally durable restorations.

Further studies with larger sample sizes and longer follow-up periods are needed to confirm the effectiveness of the connective tissue grafting technique in combination with EDTA gel for root surface cleaning. Additionally, it would be valuable to compare the outcomes of this technique with other periodontal procedures to determine the most effective and efficient treatment for gingival recession.

## 5. Conclusions

A case with high esthetic demands can be treated successfully by an interdisciplinary approach including connective tissue grafts, CAD/CAM veneers, and a combination of periodontal and restorative treatments.

## Figures and Tables

**Figure 1 medicina-59-00822-f001:**
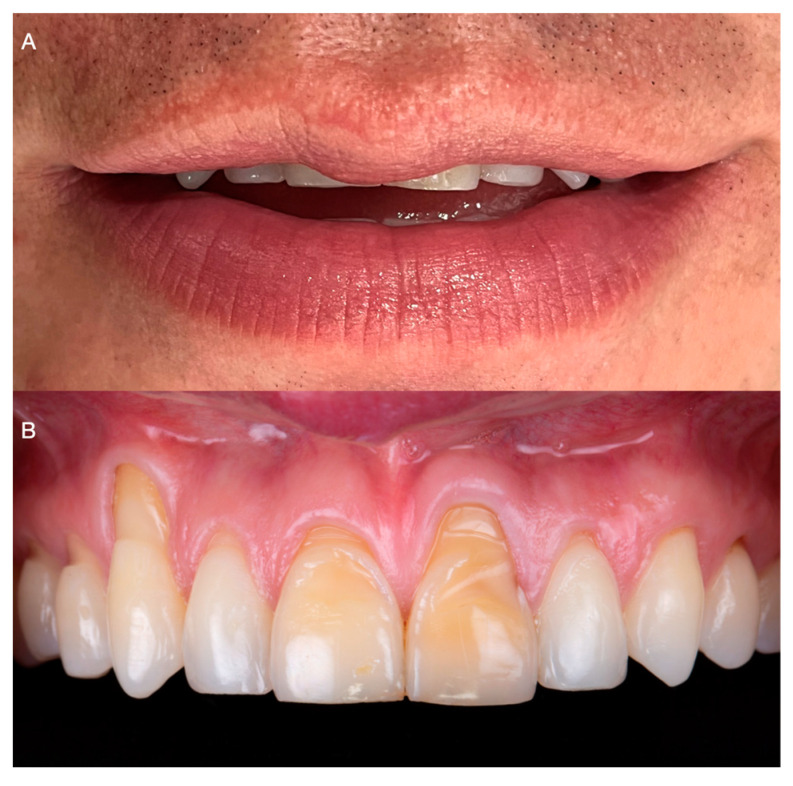
Initial situation. (**A**) Smile. (**B**) Intra-oral frontal.

**Figure 2 medicina-59-00822-f002:**
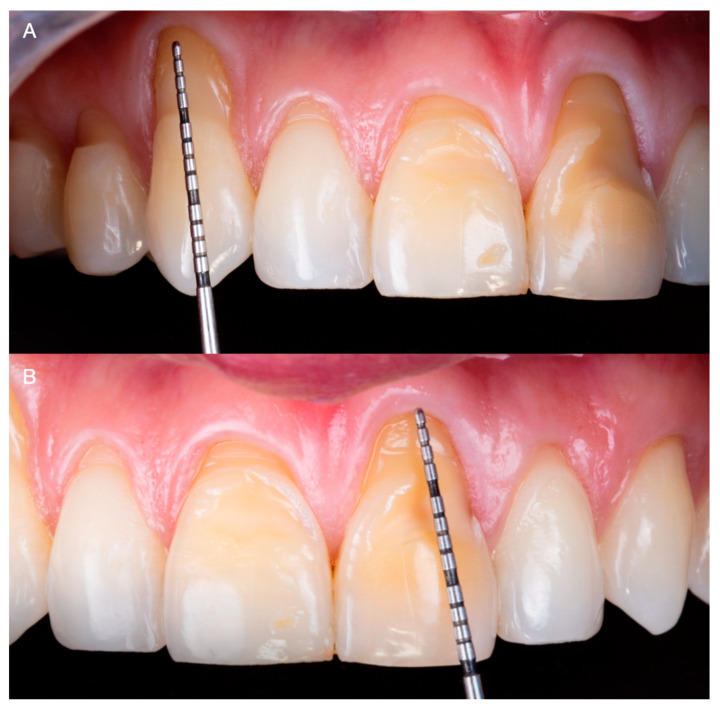
Initial intra-oral situation. (**A**) Measurement of Miller Class 1 recession on the maxillary right canine. (**B**) Measurement of Miller Class 1 recession on a maxillary left central incisor.

**Figure 3 medicina-59-00822-f003:**
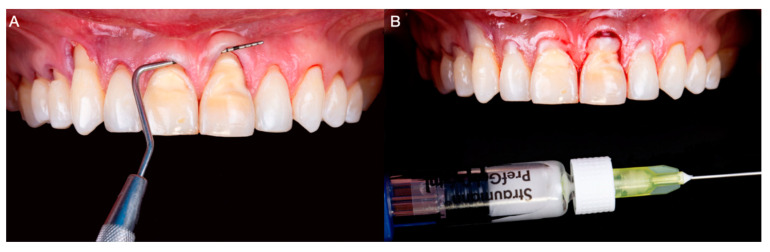
(**A**) Tunneling procedure. (**B**) 24% ethylenediaminetetraacetic acid gel application to prepare the denudated root surfaces.

**Figure 4 medicina-59-00822-f004:**
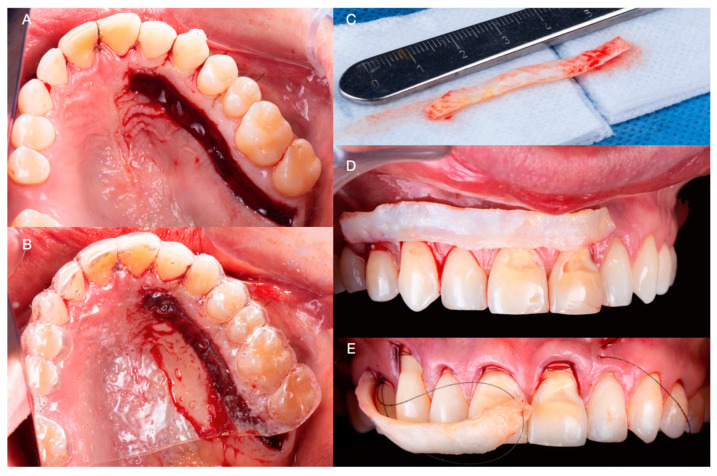
(**A**) Intraoral image of the donor site. (**B**) Donor site protected with plastic sheet. (**C**) Connective tissue graft. (**D**) Connective tissue graft over gingival recessions. (**E**) Insertion of the connective tissue graft under detached interdental papilla and marginal gingiva.

**Figure 5 medicina-59-00822-f005:**
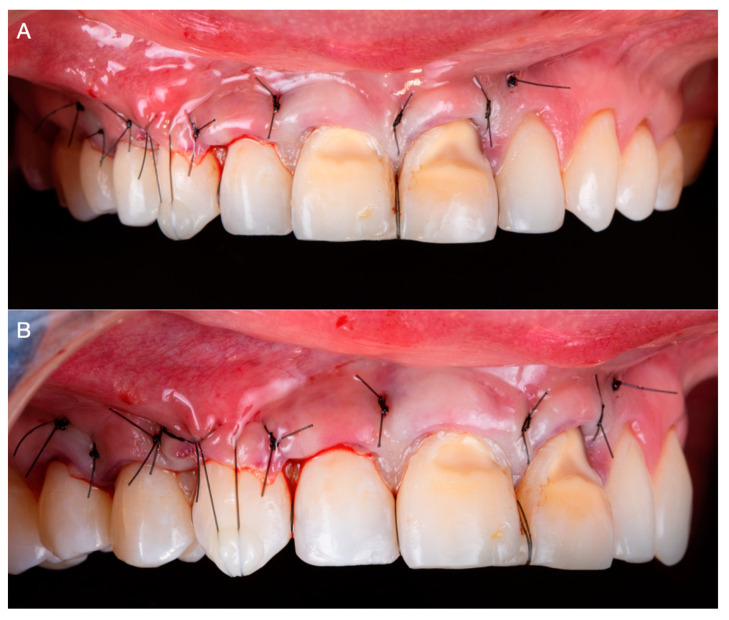
(**A**) Frontal intra-oral image after tunneling procedure. (**B**) Lateral intra-oral image after tunneling procedure.

**Figure 6 medicina-59-00822-f006:**
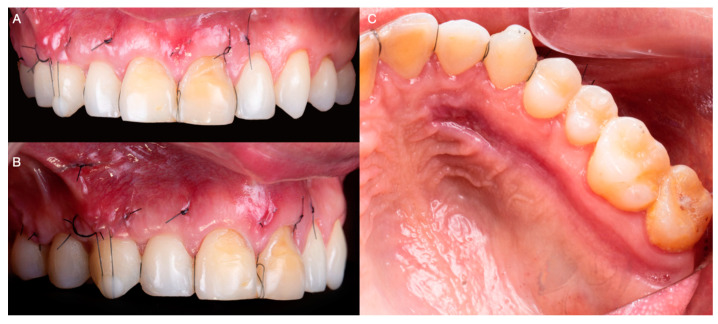
(**A**) Frontal image 1 month after surgery. (**B**) Lateral image 1 month after surgery. (**C**) Image of donor site 1 month after surgery.

**Figure 7 medicina-59-00822-f007:**
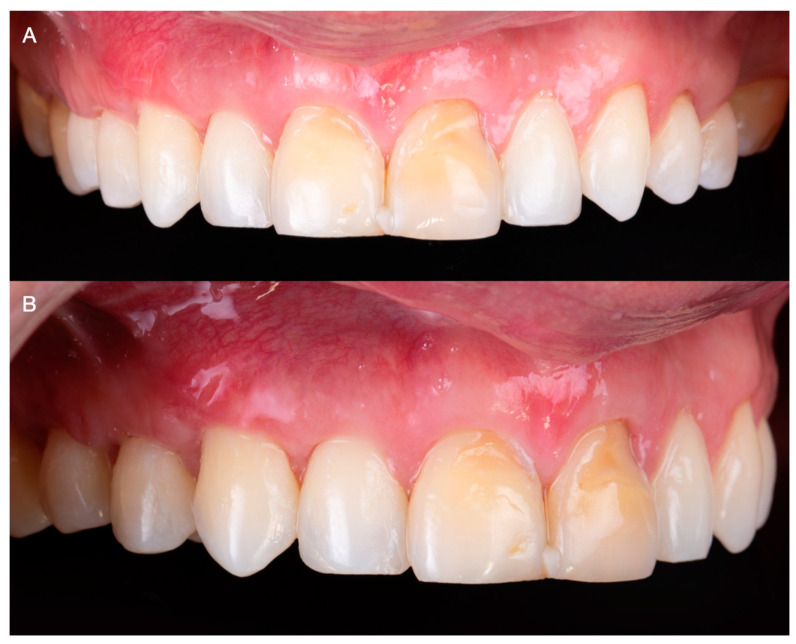
(**A**) Frontal image after 6 months of healing. (**B**) Lateral image after 6 months of healing.

**Figure 8 medicina-59-00822-f008:**
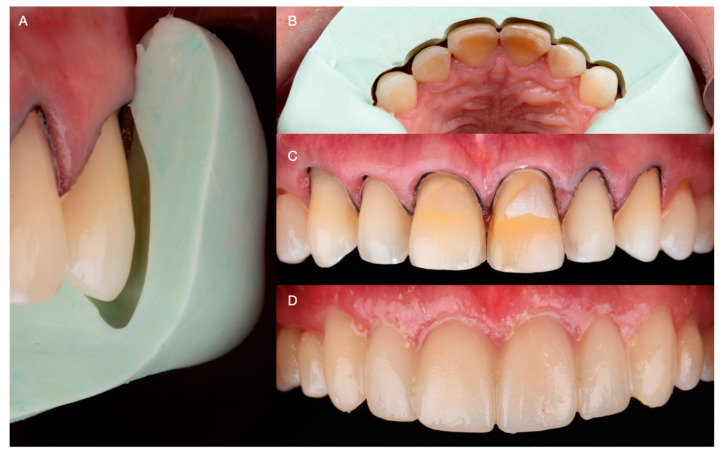
(**A**) Lateral image of minimally invasive veneer preparation with silicone reduction guide. (**B**) Occlusal image of minimally invasive veneer preparation. (**C**) Frontal image of final preparations. (**D**) Provisional restorations.

**Figure 9 medicina-59-00822-f009:**
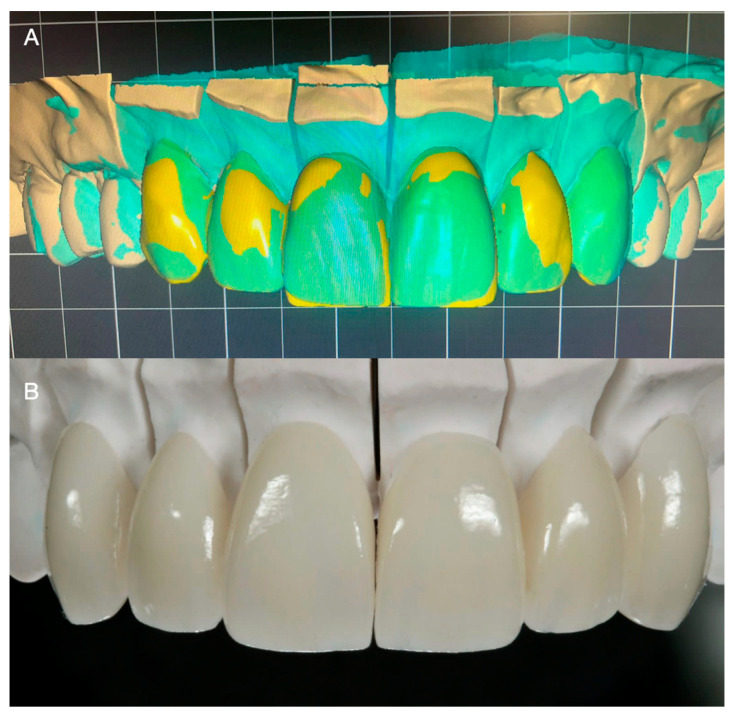
(**A**) Digital design of the veneers. (**B**) Finished ceramic veneers.

**Figure 10 medicina-59-00822-f010:**
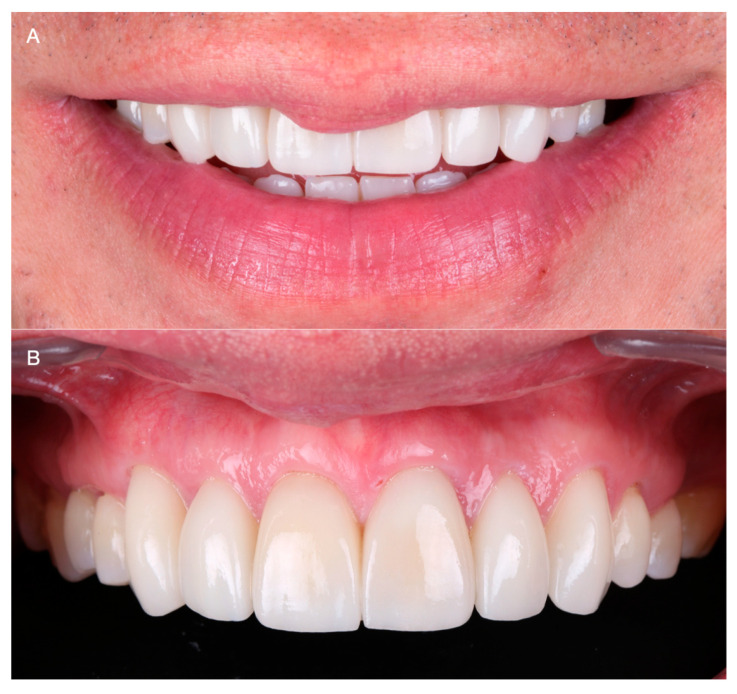
One-year follow-up. (**A**) Smile. (**B**) Intra-oral.

## Data Availability

Not applicable.

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
