# Peer review of "Soft Tissue Grafting Procedures before Restorations in the Esthetic Zone: A Minimally Invasive Interdisciplinary Case Report"

_medicina, 2023, doi:10.3390/medicina59050822_

Round 1

Reviewer 1 Report

Dear Authors, 

please justify the importance of your work- add a paragraph showing a comparison with previously done similar case reports...

figure 3c and 3d is missing 

why you have presented a case report like a research article ? -justify and change it as a case report.

you didn't mention and discuss anything about the medical history of the patient. 

introduction and discussion is weak - describe the previous case report and different treatment option for such cases in the introduction.; discuss the possible positive and negative outcomes of such cases with different treatment methods, then discuss the same for your method....

the sentence - "Further studies with larger sample sizes and longer follow-up periods are needed to confirm the effectiveness of the connective tissue grafting technique in combination with EDTA gel for root surface cleaning."- on what basis you are writing this ? have you mentioned previous studies and checked them?

Author Response

REVIEWER #1 COMMENT: Dear Authors, 

please justify the importance of your work- add a paragraph showing a comparison with previously done similar case reports...

AUTHORS ANSWER: Dear reviewer, We appreciate the time you have taken to evaluate our manuscript, and thank you for your valuable comment. In response to your suggestion, we have included a paragraph that provides a comparison with previously reported similar case reports. We believe that this addition strengthens the importance and relevance of our work.

REVIEWER #1 COMMENT: figure 3c and 3d is missing 

AUTHORS ANSWER: Thank you for your attention to detail and for pointing out the issue with Figures 3c and 3d. We apologize for any confusion this may have caused, and we have corrected the labeling in the updated version of the manuscript.

REVIEWER #1 COMMENT: why you have presented a case report like a research article ? -justify and change it as a case report.

AUTHORS ANSWER: We appreciate your comment and would like to clarify that we did not intend to present our work as a research article. We have updated the title of the manuscript to more accurately reflect that it is a case report. We appreciate your feedback, and we strive to ensure that our work is appropriately categorized and communicated to our readers.

REVIEWER #1 COMMENT: you didn't mention and discuss anything about the medical history of the patient. 

AUTHORS ANSWER: Thank you for your feedback. We agree that the medical history of the patient is important and have included new information about their medical condition in the updated version of the manuscript. We appreciate your attention to detail and your valuable comment.

REVIEWER #1 COMMENT: introduction and discussion is weak - describe the previous case report and different treatment option for such cases in the introduction.; discuss the possible positive and negative outcomes of such cases with different treatment methods, then discuss the same for your method....

AUTHORS ANSWER: We appreciate your thorough evaluation of our work and thank you for your comment. In response, we have added new information to the introduction and discussion sections to provide a more comprehensive background and context for our case report. We believe that this addition enhances the significance and impact of our work.

REVIEWER #1 COMMENT: the sentence - "Further studies with larger sample sizes and longer follow-up periods are needed to confirm the effectiveness of the connective tissue grafting technique in combination with EDTA gel for root surface cleaning."- on what basis you are writing this ? have you mentioned previous studies and checked them?

AUTHORS ANSWER: Thank you for your feedback. We have included new information in the manuscript in response to your suggestion. We value your insights and comments and appreciate the opportunity to improve our work.

Reviewer 2 Report

Dear authors,

Thank you for your great work.

This is a clinical report presents the interdisciplinary management of a 32-year-old patient with high esthetic complaints related to asymmetric anterior gingival architecture and severely discolored and eroded maxillary anterior teeth. Since the esthetically pleasing smile becomes more and more important for physical appearance and plays a significant role in daily life, the interdisciplinary management of achieving a harmonious balance between dental and soft tissue esthetics is of great value.

The manuscript has been well-written and falls within the scope of the journal. The authors have been thorough in presenting their clinical treatment procedure as well. The whole treatment procedure is also of great significance to the future clinical studies. However, some minor concerns should be addressed before this manuscript is accepted for publication.

1) The authors chose 24% EDTA gel to clean the root surface instead of the traditional compounds, please specify the specific shortage of the previous material. 

2) Ln220: Please add some evidence on how the CAD-CAM technologies make     the esthetic ceramic restorations be manufactured consistently and predictably.

3) Ln 260: While the suggestion to compare the outcomes of this technique with other periodontal procedures is valuable, the author should provide more specific information on what procedures to compare it with and why.

Author Response

REVIEWER #2 COMMENT: Dear authors,

Thank you for your great work.

AUTHORS ANSWER: We are grateful for your positive feedback and appreciate your support of our work. We strive to produce high-quality research that advances the field, and we are pleased that our manuscript has been well received.

REVIEWER #2 COMMENT: This is a clinical report presents the interdisciplinary management of a 32-year-old patient with high esthetic complaints related to asymmetric anterior gingival architecture and severely discolored and eroded maxillary anterior teeth. Since the esthetically pleasing smile becomes more and more important for physical appearance and plays a significant role in daily life, the interdisciplinary management of achieving a harmonious balance between dental and soft tissue esthetics is of great value.

The manuscript has been well-written and falls within the scope of the journal. The authors have been thorough in presenting their clinical treatment procedure as well. The whole treatment procedure is also of great significance to the future clinical studies. However, some minor concerns should be addressed before this manuscript is accepted for publication.

1) The authors chose 24% EDTA gel to clean the root surface instead of the traditional compounds, please specify the specific shortage of the previous material. 

AUTHORS ANSWER: Thank you for your comment. We have included more information regarding the choice of 24% EDTA gel for cleaning the root surface in the updated version of the manuscript. We believe that this addition clarifies the rationale and significance of our methodology.

REVIEWER #2 COMMENT: 2) Ln220: Please add some evidence on how the CAD-CAM technologies make     the esthetic ceramic restorations be manufactured consistently and predictably.

AUTHORS ANSWER: Thank you for your suggestion. We have updated the manuscript with new information and references that provide evidence for how CAD-CAM technologies can consistently and predictably manufacture esthetic ceramic restorations. We appreciate your insightful comment and value the opportunity to enhance the quality of our work.

REVIEWER #2 COMMENT: 3) Ln 260: While the suggestion to compare the outcomes of this technique with other periodontal procedures is valuable, the author should provide more specific information on what procedures to compare it with and why.

AUTHORS ANSWER: Thank you for your feedback. We have included more specific information in the discussion section to compare the outcomes of our technique with other periodontal procedures, as you suggested. We appreciate your valuable comment and are committed to improving the clarity and impact of our manuscript.

Round 2

Reviewer 1 Report

congratulations

Author Response

Dear Reviewer,

Thank you for taking the time to review our work. We greatly appreciate it.

Sincerely,

Authors